

# There's the Rub: a narrative review of the benefits and complications associated with Vicks VapoRub use

Austin Valido[1], Ana Carolina Boncompagni[1], Michelle Tsang[2] and Patricia Hume[2]

[1] Stanford University School of Medicine, Stanford, CA, United States of America
[2] San Francisco Free Clinic, San Francisco, CA, United States of America

## ABSTRACT

**Background.** Vicks VapoRub (VVR) is a globally popular over-the-counter remedy marketed for use as a topical antitussive and analgesic. This review seeks to provide health professionals and care providers with a thorough summary of the benefits and complications associated with VVR use reported in the medical literature, identify off-label consumer behaviors that might increase the risk of health complications, and encourage further research into over-the-counter (OTC) medications.

**Methods.** Three databases—PubMed, Web of Science, and Embase—aided in creating a pool of 220 studies. For inclusion, studies had to discuss the therapeutic application of VVR in humans or a complication arising from its use.

**Findings.** Thirty-seven articles were found to meet inclusion criteria. Nine synthesis groups were created: three groups concern the product's efficacy in regard to upper respiratory tract infection (URTI) symptom treatment ($n = 7$), onychomycosis treatment ($n = 2$), and in the context of wound care ($n = 2$). Six groups collate case reports that describe VVR-related complications, including VVR-associated lipid pneumonia ($n = 11$), ocular injury ($n = 5$), camphor toxicity ($n = 5$), dermatological conditions ($n = 3$), psychosis ($n = 1$), and respiratory distress ($n = 1$). The evidence around the use of VVR for the treatment of URTI symptoms and onychomycosis is limited and inconclusive. Even more limited cross-sectional data concerns its use in wound care. Many of the complications described in our study (*e.g.,* multi-organ failure, ocular evisceration, severe pneumonia) involve product misuse. This review was designed to help guide patients on the safe and appropriate use of VVR (*e.g.,* advising patients to avoid heating, ingesting, or intranasal/intrabuccal/ocular application of the product). In general, the severity of complications associated with misuse of VVR highlights the importance of complementary and alternative medicine disclosure and discussion.

# INTRODUCTION

In 1894, Lunsford Richardson Jr., a Greensboro pharmacist, created the product that would become Vicks VapoRub (VVR). Today, VVR—with its active ingredients of camphor, menthol, and eucalyptol in a petrolatum base—is marketed as a topical antitussive and analgesic, to be used with minor upper respiratory tract infections (URTI) or muscle and joint aches. A century of advertising and popular culture has established VVR's status as

Corresponding author
Austin Valido, avalido@stanford.edu

a cure-all for many people worldwide, with advertising and memorabilia dedicated to the balm (*Bermudez, 2019*). However, the product's ubiquity is only hinted at in the medical literature. Domestically, a 2015 survey of adults over the age of 65 in the US ($n = 60$) found VVR used as a home remedy in 45% of the study population (*Quandt et al., 2015*). Internationally, a cross-sectional survey of Mexican parents with young children found 61.3% had self-medicated with VVR (*Alonso-Castro et al., 2022*). There are additional small sample surveys that show Vicks products being used by the Kohaga tribe of southwest India and by mothers in Cambodia (*Bazzano et al., 2017*; *Margaret & Shetty, 2016*). A 1990 *Los Angeles Times* article describes Vicks products as being sold in over 100 countries, with 100 million gallons of VVR sold globally (*Associated Press, 1990*).

Since 1972, the product's active ingredients have been regulated under the Food and Drug Administration (FDA) monograph process, which allows certain ingredients, at certain doses, to bypass the current standard process of evaluating a drug with clinical safety and efficacy studies and be labeled generally recognized as safe and effective (GRASE) (*Bodie, 2021*; *Blake & Raissy, 2016*). The unique regulatory place of many over-the-counter (OTC) medications is reflected in the debate around VVR in the medical literature: In 1994, the American Association of Pediatrics declared that alternative therapies to camphor-containing products be considered in treating children with cough, citing the 33 pediatric cases of life-threatening camphor toxicity reported to the American Association of Poison Control Centers from 1985 through 1989 (*Committee on Drugs, 1994*). Nineteen years later, in April 2013, the AAP retired the statement (*Committee on Drugs, 1994*).

In light of this history, we hope this narrative review will provide health professionals and care providers with a thorough summary of the reported benefits and complications associated with VVR use in the medical literature, spread awareness of off-label consumer behaviors that might increase the risk of health complications, and encourage further research into OTC medication misuse.

## SURVEY METHODOLOGY

Three databases, PubMed, Embase, and Web of Science, were queried from their respective database inception to December 12, 2023 with the search terms: (Vicks VapoRub) OR (Camphor AND Menthol AND Eucalyptus). Due to a limited evidence base, we elected to not constrain the time frame of our search. For inclusion, article abstracts must have reported an analysis of the potential benefits or complications from the therapeutic use of VVR on human subjects. Studies that analyzed unnamed products with the exact composition of active ingredients and in the same medium as VVR are also included. Studies that involved non-human use, use outside of the medical context, molecular mechanisms, sociological analyses, or an analysis of any single ingredient of VVR alone were excluded. Abstract exclusion was mediated by a single reviewer (AAV) based on the criteria listed above. Full-text review of the remaining articles were examined for eligibility, using the same criteria as the abstract review stage, by two reviewers (AAV, ACB) (see Fig. 1).

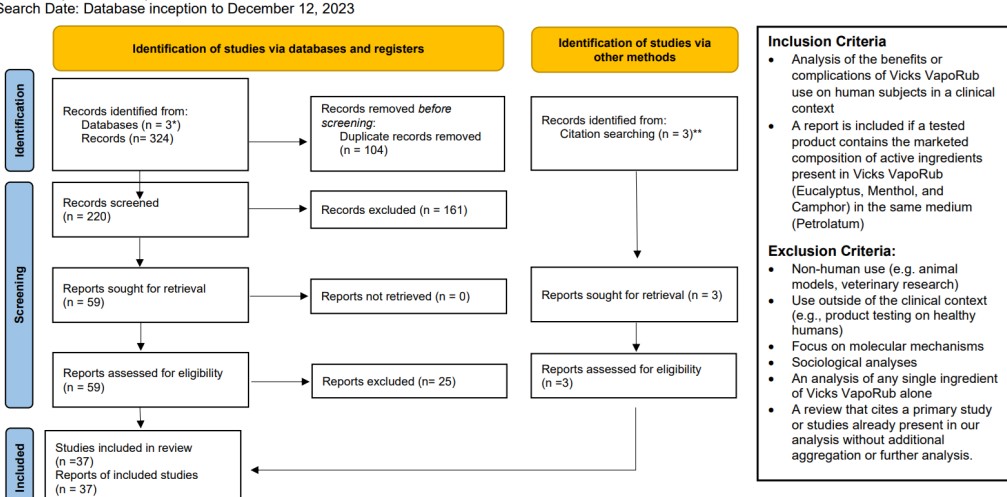

**Figure 1** **PRIMSA 2020 flow diagram.** * PubMed, Embase, Web of Science. ** Review of the citations of *Smith & Matthews (2022)* found two studies not identified from the database search that met inclusion criteria (*Eccles et al., 2015*; *Santhi et al., 2017*). Review of *Ruha, Graeme & Field (2003)* revealed an additional case study of Vicks VapoRub associated with camphor poisoning that was found to meet inclusion criteria (*Gouin & Patel, 1996*). From: *Page et al. (2021)*. For more information, visit: http://www.prisma-statement.org/.

## RESULTS

From the initial pool of 220 articles, abstract screening excluded 161 papers. Fifty-nine ($n = 59$) articles were included for full-text review. Examining the citations of a review by *Smith & Matthews (2022)*, we found two additional studies not identified by database search that met inclusion criteria ($n = 2$) (*Eccles et al., 2015*; *Santhi et al., 2017*; *Smith & Matthews, 2022*). In addition, review of the case report from *Ruha, Graeme & Field (2003)* revealed an additional case study of VVR use associated with camphor poisoning that was found to meet inclusion criteria ($n = 1$) (*Gouin & Patel, 1996*; *Ruha, Graeme & Field, 2003*). Twenty-five ($n = 25$) articles were further removed, including review articles that solely cite sources otherwise included as a primary study ($n = 12$), reports with full-text unavailable in English ($n = 4$), articles published as a comment to another publication with no additional analysis ($n = 2$), articles that were found to interrogate a single ingredient of VVR or did not contain the same active ingredients as VVR ($n = 7$). In total, the full-text of 37 articles were analyzed for this review.

The 37 articles were thematically separated into nine synthesis groups. Of note, the analysis groups are separated broadly as relating to efficacy (relating to treatment of a medical condition) and complications despite these being intrinsically related categories (*e.g.*, patients developing a complication after use to treat cold symptoms). We found this separation allows for a more complete picture of the complications associated with VVR use by the general public, with all varieties of duration of use and off-label use. With that in mind, three groups look at the efficacy of VVR for the improvement of URTI symptoms

($n = 7$), onychomycosis treatment ($n = 2$), and pruritus treatment/wound care ($n = 2$). Six additional groupings are clustered around case reports of complications, including lipid pneumonia ($n = 11$), camphor toxicity ($n = 5$), contact dermatitis and leukoderma ($n = 3$), ocular injury ($n = 5$), psychosis ($n = 1$), and respiratory distress ($n = 1$).

## Efficacy

### *Upper respiratory tract infection symptom treatment*

In the 1970s, Berger et al. first interrogated the clinical effects of VVR in pediatric patients with acute bronchitis using objective measures (*Berger, Jarosch & Madreiter, 1978*; *Berger et al., 1978*; *Berger, Madreiter & Jarosch, 1978*). However, the lack of blinding and clinically-significant outcome measures prevents any useful interpretations of their studies. Modern studies continue to struggle to properly analyze or measure the clinical effects of Vicks VapoRub for its most common use—treatment of URTI symptoms. In terms of higher-level evidence, there have been three randomized control trials (RCTs) that focused on the use of VVR to alleviate symptoms of the common cold (*Eccles et al., 2015*; *Paul et al., 2010*; *Santhi et al., 2017*).

The sole RCT involving pediatric patients ($n = 138$) found the treatment group to have statistically significant improvement in parent-reported perceived quality of sleep, for both the child ($p = .006$) and the parent ($p = .008$), and improvement in an overall combined symptom score ($p = .03$) when compared to a petrolatum control (*Paul et al., 2010*). The authors fail to assign a primary endpoint. With Bonferroni correction, only one of the seven outcomes reaches statistical significance: the unblinded parent reporting their child's sleep quality (corrected $p = .007$). In addition, the study also finds a statistically significant presence of adverse events associated with the treatment group with mild skin ($p < .001$), eye ($p < .001$), and nose irritation ($p < .001$) (*Paul et al., 2010*). Two studies focused on adult patients. *Eccles et al. (2015)* found the VVR treatment groups to have a significant decrease in subjective time to both the first sensation of nasal cooling and the first sensation of nasal decongestion, a clinically meaningless trial and outcome. A second study assessed their primary outcome through a sleep survey (the SQSQ), which had a patient grade sleep quality after awakening, and found a significant, but clinically uninterpretable, improvement of 7.9 (SE 3.78) points between experimental groups on a 100-point scale (*Santhi et al., 2017*). *Santhi et al. (2017)* fails to cite an accurate source for their validated sleep survey, citing three studies with three different measurement tools, the most similar being unvalidated. Despite separately analyzing 24 additional endpoints, no corrections for multiplicity were conducted. Objective measures captured by sleep actigraphy failed to reject the null hypothesis. All other survey questions, the Karolinska Sleep Diary and an additional researcher-created survey, also fail to reject the null hypothesis with a Bonferroni correction. Due to the nature of the treatment, all three studies struggled to blind patients to their group (*Paul et al., 2010*; *Eccles et al., 2015*; *Santhi et al., 2017*). In addition, all three of these studies were funded by Procter and Gamble.

No clinical evidence exists for the use of VVR in any other respiratory condition. A 2014 cross-sectional study of a Bronx high-school found the use of VVR, after controlling for severity, to be associated with lower asthma morbidity, defined in this study by

emergency department use (*Reznik, Sharif & Ozuah, 2004*). The correlational study's conclusion ignores the more likely socioeconomic divisions between the two experimental groups—children either treated with a clinician-prescribed medication or an inexpensive over-the-counter medication, namely the social determinants of healthcare access, and how that might affect emergency department access and use.

### Onychomycosis treatment

Two open-label single-arm clinical trials published in the literature concern the use of VVR as a treatment for onychomycosis (*Derby et al., 2011*; *Snell et al., 2016*). In 2011, Derby et al. analyzed 18 patients that completed a 48-week treatment of a single, daily application of VVR to affected nails. The study found five of the 18 patients (27.8%) had a mycological cure (defined as a negative nail culture) and 10 (55.6%) showed a decrease in area of the dystrophic nail. The average ratio of affected to total nail area, based on manually measured photographs, decreased from 63% at initial evaluation to 41% at 48 weeks ($p < .001$; paired $t$ test) (*Derby et al., 2011*). Five years later, *Snell et al. (2016)* conducted a single-site, prospective pilot study of VVR to treat onychomycosis in patients living with HIV ($n = 20$, with 18 patients seen at the final 24-week timepoint). After 24-weeks, the study reported 83% of the study population showing improvement of the affected nail (median clearance 25%, range 6.3% to 87.5%). The lack of a control group and methodological detail with both reports prevents any current clinical recommendations around this off-label use.

### Pruritus treatment and wound care

Only cross-sectional and survey-based studies were found to engage on the topic of VVR— or as described in the *Danial et al. (2015)* study a "vaporizing rub (menthol, camphor, and eucalyptus)"—and its use in the context of wound care. A 2023 analysis of the patient perspectives of wound care management in hidradenitis suppurativa ($n = 302$) found VVR used by 36.8% of the study population ($n = 111$ patients) (*Poondru, Scott & Riley, 2023*). Within this population, VVR received the highest proportion of users rating the product efficacy as good or excellent for treatment of acute nodules (59.5%, $n = 66$, $p < 0.01$). Likewise, a descriptive 2014 survey of patients with epidermolysis bullosa and their caregivers ($n = 146$) found 16 of the participants using a 'vaporizing rub', with an average 'effect on pruritus' score of $-0.9$ on a five-point scale ($-2$ to $+2$) (*Danial et al., 2015*). The lack of any casual evidence of benefit, as well as trial data showing the significant presence of skin irritation in children, prevent any current clinical recommendations around this off-label use.

## Complications
### Lipid pneumonia

First defined in 1925, lipid pneumonia occurs when lipid-based products, such as petrolatum, accumulate in the alveoli and distal airways, leading to an inflammatory response that decreases the efficiency of gas exchange (*Beck & Landsberg, 2024*). VVR-associated lipid pneumonia has been documented in 11 case reports (*Cabri et al., 2017*; *Cherrez Ojeda et al., 2016*; *Cohen & Schoene, 1963*; *Gattuso, Reddy & Castelli, 1991*; *Kamal, Azzi & Elsayegh, 2015*; *Kilaru et al., 2017*; *Kulkarni & Pena, 2012*; *Martin, 2013*; *Rangarajan*

*et al., 2012; Sargi et al., 2021; Subramanian et al., 1982*). Case reports mostly describe the placement of the product around the nostrils at least several times a week for over a year (*Cabri et al., 2017; Cherrez Ojeda et al., 2016; Kamal, Azzi & Elsayegh, 2015; Kilaru et al., 2017; Kulkarni & Pena, 2012; Martin, 2013; Rangarajan et al., 2012; Subramanian et al., 1982*). Diagnostic measures including fine needle aspiration biopsies, bronchoalveolar lavages and lobectomies. *Cherrez Ojeda et al. (2016)* proposed that high-resolution computed tomography (CT) is the best imaging modality for diagnosis of this disease, although a definitive diagnosis requires invasive biopsy techniques of lung parenchyma (*Beck & Landsberg, 2024*). Histopathological evidence of lipid-laden macrophages in respiratory samples confirms the diagnosis (*Beck & Landsberg, 2024*). Treatment includes the cessation of the offending agent and some case reports describe improvement of symptoms with corticosteroids (see Table 1).

## Camphor poisoning

In 2017, *Torres et al. (2018)* described the case of a 54-year old woman admitted to the ICU secondary to a presentation of multiorgan failure, with labs significant for hypoxemia, respiratory and metabolic acidosis, elevated creatinine, and elevated liver enzymes. The patient was intubated, resuscitated with fluids, and given empiric antibiotics. Upon discovery that the patient ingested three 50 g containers of VVR on a weekly basis, and consultation with poison control, a diagnosis of camphor intoxication was made. The patient improved with discontinuation of camphor-containing products and was discharged after 10 days.

Case reports involving the ingestion of VVR more frequently involve children and have been reported in the medical literature for decades, ranging in time from 1976 to 2019 (*Boland, Stevens & Gosen, 2019; Gouin & Patel, 1996; Phelan, 1976; Ruha, Graeme & Field, 2003*). The above case reports show a wide-range of the development of camphor-toxicity related symptoms, from 2 hours to 9 hours after ingestion, a wider range than the 90-minute expectation timeline reported multiple times in the literature (*Boland, Stevens & Gosen, 2019; Gouin & Patel, 1996; Ruha, Graeme & Field, 2003*). *Torres et al. (2018)* notes there is no correlation between patient symptomatology, prognosis, and camphor levels in the blood. Current treatment recommendations are supportive care with activated charcoal and gastric lavage as potential means for detoxification, benzodiazepines for seizure treatment, and intubation for airway protection if needed.

## Ocular injury

The *Paul et al. (2010)* trial reported the statistically significant adverse effect of "burning sensation of the eyes" (16% of the treatment group, $P < .001$). Four additional case reports detail ocular injuries sustained as a direct result of the use or misuse of VVR (*Brazda, 2004; Fung & Oxford, 2004; Gagnon & Walter, 2004; Sahay et al., 2017*). Two of these case reports included patients who inadvertently superheated Vick's VapoRub in the microwave, leading to an explosion directly into their eyes while attempting to inhale the heat vapor (*Fung & Oxford, 2004; Gagnon & Walter, 2004*). The *Gagnon & Walter (2004)* report patient, a 40-year-old male patient with an almost total corneal epithelial defect, failed to heal his injuries secondary to limbal stem cell injury.

**Table 1  Compiled case reports of lipid pneumonia secondary to Vicks VapoRub use.**

| Author, Year | Description of patient use | Symptomatology | Imaging | Biopsy | Treatment | Naranjo score |
|---|---|---|---|---|---|---|
| *Cohen & Schoene, 1963* | Three years of putting a tablespoon of product in the back of his throat each night for dry throat relief | 55-yo white male with history of smoking, presented with one year of dry cough and intermittent substernal chest pain | CXR | Scalene biopsy, Right lower lobectomy | No treatment reported | 4 |
| *Subramanian et al., 1982* | Five years of use in and around nostrils for sinus troubles | 61-yo white male farmer, presented with fever, chills, fungal colonization | CXR | Fiberoptic bronchoscopy, transbronchial biopsy | Cessation | 4 |
| *Gattuso, Reddy & Castelli, 1991* | Inhaling for multiples years to relieve asthma symptoms | 70-yo female with history of asthma, presented with 10-month history of SOB, hoarseness, and decreased exercise tolerance | CXR, CT | CT-guided FNA biopsy | No treatment reported | 4 |
| *Kulkarni & Pena, 2012* | Regular use at night by applying it inside nose several times per week for years | 68-yo white male with history of moderate COPD, 80 pack year smoking history, presented with minimal shortness of breath then, after being lost to follow up, returned a year later with worse dyspnea and cough | CXR, CT, PET | Left upper lobectomy, hilar lymph node resection | Cessation | 4 |
| *Rangarajan et al., 2012* | Inside nostrils at bedtime for several years | 40-yo white female presented with shortness of breath on exertion for two years | HRCT | VATS lung biopsy | Cessation(Previously trialed on 1 year of steroids and azathioprine) | 4 |
| *Martin, 2013* | History of use in her nostrils daily as a child and current mineral oil use for constipation | 49-yo female presented with cough and dyspnea on exertion for several months became oxygen dependent and resulted in super infection | PET | CT-guided biopsy, open lung biopsy | Cessation, whole lung lavage, lung transplant clinic referral | 0 |
| *Kamal, Azzi & El-sayegh, 2015* | Twenty years applied intranasally to relieve congestion | 55-yo male with history COPD, increased cough for 2 years, 93% saturation on room air | CT | Transthoracic FNA | Cessation | 3 |
| *Cherrez Ojeda et al., 2016* | Daily use for 50 years onto chest, palms, feet and aspirated through nose | 85-yo female patient with history of allergic rhinitis presenting without respiratory symptoms for routine annual physical exam with elevated CRP | CXR, CT | Thoracic CT density values indicating lipoid mass with improved follow up scan ruling out malignancy | Cessation, intranasal corticosteroids | 4 |
| *Kilaru et al., 2017* | Over a year of external nostril application nightly | 23-yo female with 5 year history of chronic rhino-bronchial allergies, presented with cough, dyspnea, and fever for >4 weeks | CXR, HRCT | Fiberoptic bronchoscopy, transbronchial lung biopsy | Oral prednisone for 6 months, continued inhaled corticosteroid and Fluticasone spray | 6 |
| *Cabri et al., 2017* | Daily intranasal application for over 4 months | 57-yo female presented with dry cough over 4 months | CT | Computed tomography (CT) of the chest showed foci of consolidation and ground-glass attenuation in gravity-dependent segments of the right lower lobe. | Cessation | 4 |
| *Sargi et al., 2021* | Continuous use for many years | 79-yo female who presented with non-productive cough and dyspnea on exertion for over a month, requiring supplemental oxygen. | CXR, CT | Bronchoalveolar lavage sample | Steroids without improvement, No other treatment reported | 2 |

Inflammatory reactions from direct placement on the eyelids or face are detailed in two case reports. *Sahay et al. (2017)* presents a case of a 40-year-old woman with a 15-day history of headache, and the acute onset of bilateral eye pain, redness, and discharge. The patient's visual acuity had diminished to the point of only perceiving light, in addition to severe corneal edema and epithelial defects in both eyes. Upon further history-taking, and excluding other possible diagnoses, the patient was determined to

have acute keratitis due to a VVR chemical injury. She reportedly was not able to fully recover her baseline vision. In addition, *Jaiwal (1989)* reports a 54-year old man's case of combined dermatitis/keratoconjunctivitis after application of VVR to the forehead, eye area, and nose the night prior for cold relief. The patient presented to clinic with bilateral vesicular eruptions and ulcerations of the eyelids, along with bilateral clouding and epithelial irregularities of the corneas. The patient was treated with mydriatic drops; topical, subconjunctival, and systemic antibiotics; and acetazolamide. The patient's right required evisceration, completed at a later date, with visual acuity of the left improving slightly despite residual corneal clouding (*Jaiwal, 1989*).

The final three case reports present a different mechanism of ocular injury: application on the ocular surface through aerosolized spray or direct application. *Brazda (2004)* describes an 18-year-old patient who required medical attention due to bilateral eye pain after a companion blew VVR through an inhaler directly into the eyes of a person under the influence of MDMA, a popular act. This patient was described to have clouding of corneas with evidence of bilateral corneal abrasions on fluorescein stain. His care was transferred to ophthalmology, and he was prescribed prednisolone eye drops in addition to pain medication.

### Contact dermatitis and chemical leukoderma

The 2010 *Paul et al. (2010)* trial reported some mild skin irritant effects from VVR treatment, with statistically significant reporting of burning sensation of the skin (28% of the treatment group, $p < .001$), burning sensation of the nose (14%, $p < .001$). In addition, there have been three case reports describing dermatological reactions in association with VVR. Contact dermatitis attributed to VVR use appeared in two case reports: *Noiles & Pratt (2010)* published a case report concerning a 66-year old with a history of acute contact dermatitis on her neck and anterior chest after VVR use. Patch-testing results at 120 h revealed a 3+ reaction to VVR, as well as neomycin, nickel, and dibucaine (*Noiles & Pratt, 2010*). *Mendez & Nazario (2013)* reported the case of a 47-year old woman who presented for the evaluation of chronic nasal and facial edema, erythema, and tenderness that had persisted for 1-year. Patch-testing to 50 allergies revealed a reaction to VVR, with symptoms resolving after discontinuation of the product (*Mendez & Nazario, 2013*).

In addition, a 2008 case report described chemical leukoderma associated with VVR use (*Boyse & Zirwas, 2008*). A 78-year old woman presented for evaluation after newly acquired hypopigmentation around perinasal and perioral skin. Caretakers revealed several consecutive days of VVR application in the same areas. The patient declined any treatment or further testing.

### Psychosis

A single 1982 case report described a 41-year-old woman who suffered from several months of paranoia, personality changes, and acute delirium that resolved with the cessation of VVR and Vicks Sinex Nasal Spray misuse (*Blackwood, 1982*). The patient presented to the emergency department in an acutely delirious state, with further interview and testing revealing no evidence of drug or alcohol use, except for longstanding use of over a bottle of VVR per week, placing the ointment product inside her nose, as well

as longstanding use of Vicks Sinex Nasal Spray (active ingredient, oxymetazoline HCL 0.05%). This patient started the use of both products three years prior to presentation, and progressively increased her use throughout the years. It is unclear which of the active ingredients consumed—oxymetazoline, camphor, menthol, or eucalyptol—contributed to her psychosis.

### Respiratory distress

*Abanses, Arima & Rubin (2009)* reported an 18-month previously healthy girl brought into the emergency department with severe respiratory distress, evidenced by wheezing, intercostal retractions, and initial pulse oximetry saturation of 66% on ambient air. Albuterol and prednisolone were provided without a clinical response. Supplemental oxygen increased the patient's oxygen saturation to 93%. A chest radiograph showed mild peribronchial cuffing but no infiltrates or hyperinflation. The patient was admitted to the hospital on supplemental oxygen and was discharged the following day. The patient's grandparents reported respiratory distress developing over a period of 30 to 45 min after the application of VVR.

## CONCLUSIONS

To our knowledge, this report is the first dedicated review of the benefits and complications associated with the globally popular OTC remedy Vicks VapoRub. Overall, we found only limited and inconclusive evidence supporting the clinical use of the product. The significant results of the two modern RCTs concerning clinically-relevant outcomes from the use of VVR to treat URTI symptoms, *Paul et al. (2010)* and *Santhi et al. (2017)* were statistically flawed and largely limited to subjective, self-reported improvement of sleep scores, in pediatric and adult populations respectively, with no changes in objective measures. While the other reports that utilize single-arm or survey-based methodologies report limited clinical benefit, the lack of a comparison group prevents any conclusion that VVR or generic versions of the product are efficacious or beneficial.

There is a similar paucity of research surrounding complications from VVR use, which we attribute to the product's innocuous nature when used correctly. *Paul et al. (2010)* trial documents the statistically significant adverse event of mild skin and eye irritation with a single night's use. An additional twenty-six case reports provide limited clinical evidence around complications, with specific off-label behaviors seemingly increasing the risk of dangerous outcomes, such as intranasal and intrabuccal application and occurrences of lipid pneumonia, or heating the product being associated with ocular injury. The case of multi-organ failure precipitated by camphor ingestion illustrates how uncommon, life-threatening consequences can arise when using VVR improperly (*Torres et al., 2018*). The most serious case reports—fatal lung infection, multi-organ failure—are associated with misuse.

Our study has the following limitations: only one reviewer conducted the initial abstract review, opening up the possibility of bias in the review process. However, we set firm inclusion and exclusion criteria that dictated which reports were included in our final analysis. In the future, specific use cases of VVR can be analyzed using a systematic review

process, with multiple reviewers, that further improves upon our research methodology. An additional limitation of our study is the exclusion of grey literature (*i.e.,* government documents) which could have provided nontraditional sources of information. In terms of future directions, we hope this report encourages further research on the clinical evidence around VVR use, with systematic reviews dedicated to discrete use cases (*e.g.*, URTI symptom treatment). An additional direction for future research could be an analysis of poison control center databases for additional information about the current state of camphor-related poisonings attributable to VVR or related products. Overall, we hope this article encourages healthcare practitioners to think about ways to improve the full disclosure of alternative therapies during patient visits.

## ACKNOWLEDGEMENTS

We thank Amanda Woodward, MLIS for her help in the development of our search terms.

### Funding
The authors received no funding for this work.

### Competing Interests
The authors declare there are no competing interests.

### Author Contributions
- Austin Valido conceived and designed the experiments, performed the experiments, analyzed the data, prepared figures and/or tables, authored or reviewed drafts of the article, and approved the final draft.
- Ana Carolina Boncompagni analyzed the data, prepared figures and/or tables, authored or reviewed drafts of the article, and approved the final draft.
- Michelle Tsang analyzed the data, authored or reviewed drafts of the article, and approved the final draft.
- Patricia Hume analyzed the data, authored or reviewed drafts of the article, and approved the final draft.

### Data Availability
  This is a literature review.

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
