# Peer review of "There’s the Rub: a narrative review of the benefits and complications associated with Vicks VapoRub use"

_PeerJ, doi:10.7717/peerj.19105_

## Round 0.1 · original submission · Major Revisions

Three reviewers have now commented - please address all their items.

Reviewer 1 ·

Basic reporting

The review article on Vicks VapoRub is a decent review of a top-rated OTC product in South America and the Asian subcontinent.

Experimental design

Sufficient number of studies were considered to make this review substantial

Validity of the findings

Statistical significant results are described.

Additional comments

This is an exciting review article that I enjoyed reading. Vicks VapoRub is a very popular product among South American households and in many Asian countries.
My suggestion would also include other products with the same/similar composition that are not manufactured by Proctor and Gamble, such as Vicks VapoRub (brand name). Many try to save money by not buying the brand name but the equivalent product.

Reviewer 2 ·

Basic reporting

Author credentials not listed
Need list of abbreviations
Background paragraph of abstract - change interested parties to This article seeks to provide health professionals and care providers with a thorough...
Methods paragraph of abstract - list the three databases
Findings paragraph in abstract - do not state We hope this review. Change to This review was designed to help guide patients on the safe and appropriate use of VVR.
line 49, changed cemented to established
line 50, memes - change to advertising

Experimental design

To suggest the literature search was limited to one single data is Dec 2023 is not accurate
You need to list date of inception for each data base to the search date Dec 12, 2023
PubMed inception 1966 -
Embase inception 1947 -
Web of Science inception 1900 -
Why the delay in article submission if the search was completed nearly one year ago
Your search strategy should include VVR and not unnamed products - how do you know the composition and active ingredients were identical to VVR ?
Line 87, due to time limitations - do not state this
Line 88 if only one author evaluated article exclusion - how were conflicts or uncertainties handled?
line 93 - was the Prisma flow diagram registered?
line 125 change to pediatric patients with acute bronchitis
Be consistent listcitations - see line 145 Santhi et al and Eccles 2015
Line 160 ED used as abbreviation with no definition of abbreviation

Validity of the findings

line 244 while attempting
line 276 and 277 unnecessary use of colon
Did you try to obtain any data from poison control center?
Table 1 - can you include the Naranjo score for each reported reaction?
Figure 1 major inconsistency noted. Inclusion lists equivalent active ingredients of VVR and exclusion lists a product that is not VVR. Technically if the product is not labeled as VVR, it is not VVR.
Conclusion - need significant wordsmithing. Need to be clear and concise. You should not expect to find an abundance of publications where VVR was used incorrectly or associated with an undesired outcome. Did you check poison control data bases for more data on drug misadventures?

Reviewer 3 ·

Basic reporting

Article uses professional English. However, while the article does a good job of describing efficacy and adverse events, as described in the title, it is less clear in answering its objectives which are to show the "benefits and complications" of Vicks Vaporub use. Please highlight what are the benefits and complications of Vicks Vaporub use to answer your objectives as "benefits" are different from "efficacy" and "complications" are different from "adverse events".

Experimental design

Methodology is rigorous but review does not include grey literature which could have contained additional information on benefits and complications. Please consider expanding databases to ones that include grey literature.

Validity of the findings

Findings are valid; however, it may not be representative because gray literature was not included.

Additional comments

Please add a limitations (sub)section. Also add a section on further research. Is there a way to have a graphical abstract/concise summary of the review to make it easier for laypeople who might read the manuscript?

---

## Round 0.2 · Major Revisions

You must appropriately respond in light of the comments from Reviewer 2. As you can see, they have reiterated several points and and emphasized that as the quality of evidence for each trial was extremely low, the conclusions should reflect the lack of objective efficacy data

Reviewer 1 ·

Basic reporting

The authors have addressed my concerns and the manuscript can now be accepted for publication.

Experimental design

The authors have addressed my concerns, and the manuscript can now be accepted for publication.

Validity of the findings

The authors have addressed my concerns and the manuscript can now be accepted for publication.

Additional comments

The authors have addressed my concerns, and the manuscript can now be accepted for publication.

Reviewer 2 ·

Basic reporting

Line 74, find alternative to "other interested parties"
Line 327, find alternative word for dearth

Experimental design

The authors were previously asked to not include data on unnamed products, line 86
How can you objectively ensure the active ingredients and excipients are exactly the same as VVR?
How can you objectively ensure the manufacturing process for these unnamed products were the same as VVR?
Patients can have intolerance or reactions to even a single ingredient used as excipient or preservative.

Your methods are not sound to make a generalization for conclusions, wherein not all of the data reflects VVR
You are also not able to state an overall inclusion for VVR, since you are including unnamed products which may or may not be pharmaceutically equivalent to VVR
You will need to break out data for unnamed products and comment upon any applicable conclusion for efficacy or safety

If you do not separate out the unnamed products, then you will need to make a statement regarding the generic active ingredients and not specifically the branded VVR product

Validity of the findings

The quality of evidence for each trial was extremely low
The conclusions should reflect the lack of objective efficacy data

See above concern for methodology

Additional comments

no additional comments

Reviewer 3 ·

Basic reporting

No comment.

Experimental design

No comment.

Validity of the findings

No comment.

---

## Round 0.3 · accepted · Accept

Thank you for making the requisite modifications in response to the reviewers' comments. You manuscript is now acceptable for publication.

Reviewer 2 ·

Basic reporting

wordsmithing completed

Experimental design

authors did a better job of explaining ingredients - this was an important factor that needed documentation

Validity of the findings

the quality of evidence is low - this cannot be fixed